# Are Heated Tobacco Product Users Less Likely to Quit than Cigarette Smokers? Findings from THINK (Tobacco and Health IN Korea) Study

**DOI:** 10.3390/ijerph17228622

**Published:** 2020-11-20

**Authors:** Cheol Min Lee, Choon-Young Kim, Kiheon Lee, Sungroul Kim

**Affiliations:** 1Department of Family Medicine, Healthcare System Gangnam Center, Seoul National University Hospital, Seoul 06236, Korea; bigbangx.snuh@gmail.com; 2Department of Family Medicine, Seoul National University Bundang Hospital, Seongnam 13620, Korea; omh4ever@gmail.com; 3Department of Family Medicine, Seoul National University College of Medicine, Seoul 03080, Korea; 4Department of Environmental Health Sciences, Soonchunhyang University, Asan 31538, Korea; sungroul.kim@gmail.com

**Keywords:** non-cigarette tobacco products, electronic nicotine delivery devices, cessation

## Abstract

Since the advent of heated tobacco products in June 2017 in South Korea, the sale of heated tobacco products accounted for 10.5% of total tobacco sales in 2019. However, the decreasing trend in total tobacco sales is gradually weakening and the number of visitors using stop smoking services has also dropped. This study examines the association between the use of new tobacco products and related products and cessation behaviors. A cross-sectional study using a self-administered questionnaire was conducted from March 2019 to July 2019 for 2831 adult tobacco users. The difference in rates of quit attempts using the type of tobacco products and related products in the past year were noted (55.6% (any cigarette smoker), 46.7% (any e-cigarette user), and 39.6% (any heated tobacco product user)). About a 30% increase in quit attempts was observed for the triple users of either conventional cigarette or heated tobacco product than exclusive users. Exclusive heated tobacco product and e-cigarette users were approximately 40% and 20% less likely to quit the product they used than exclusive cigarette smokers, respectively. These findings can explain recent occurrences in South Korea, such as the reduction of visitors at smoking cessation clinics and the attenuation of the decline in tobacco sales.

## 1. Introduction

Tobacco is a cardinal and preventable cause of premature death in the world, driving an epidemic of malignancy, coronary heart disease, stroke, chronic pulmonary disease, and other chronic diseases. According to the Global Burden of Disease Study, a total of 8.1 million death across the world occurred due to tobacco use in 2017 and comprising 7.1 million deaths from cigarette smoking [1]. Globally, the economic cost associated with smoking is nearly USD 2 trillion (i.e., with a 2016 purchasing power parity) each year, equivalent to almost 2% of the world’s total economic output [2]. Although abstaining from tobacco use is one of the most effective ways to save lives and improve overall well-being, less than 5% of smokers eventually succeed in quitting smoking for a year on their own [3]. For those who fail to quit, tobacco companies are always present with new tobacco products and related products. Electronic cigarettes (e-cigarettes) and heated tobacco products (HTPs) were introduced into markets previously dominated by conventional cigarettes (CCs), so smokers looking for more convenient modalities of tobacco consumption can choose such alternatives.

HTPs, also known as “heat-not-burn” products, enable processed tobacco to be heated rather than combusted in a controlled manner [4]. The tobacco company actively promoted HTPs as being odorless and less harmful [5], but this claim of tobacco company has been demonstrated in laboratory, not in real life. HTPs were launched in South Korea in June 2017 and accounted for 2.2% of the total tobacco sales in the first year of their launch, and the figure rose to 10.5% of total tobacco sales in 2019 [6,7]. Meanwhile, due to concerns from the E-cigarettes, or vaping, product use-associated lung injury (EVALI) outbreak, the Korean government encouraged e-cigarette users to stop using these products, resulting in existing vapers choosing conventional cigarettes or HTPs instead and an opportunity to expand the HTPs market. HTPs shall expand to the rest of the United States (US) within a few years if sales in the Atlanta test market go well [8,9].

While the sales of HTPs have increased, two unusual things have been observed. First, the rate of decline in total tobacco sales has slowed since HTPs was introduced (−20131.5% in 2018 and −0.7% in 2019) [10]. Second, the number of visits to smoking cessation clinics decreased by 13% and the amount of government support allocated for the prevention of smoking medication dropped 27.5% in 2018 [11,12]. These temporal changes imply that the advent of HTPs has a significant impact on tobacco controls in South Korea. Although some recent studies regarding Korean adolescents showed that the use of HTPs was associated with lower odds of abstinence from CCs [6,13], no study has confirmed the association between the utilization of HTPs and the reduction of quit attempts to our best knowledge. Considering the statistics related to tobacco sales and numbers of visitors to smoking cessation clinics, there is a possibility that people who choose HTPs are less interested in quitting tobacco from the start. In this regard, the present study examined the relationship between each type of product (CC, e-cigarette, and HTP) and its combinations (single, dual use, and triple use) and quit attempts in South Korea. We hypothesized that the current use of HTPs will be associated with fewer quit attempts. 

## 2. Materials and Methods

### 2.1. Study Participants

THINK (Tobacco and Health IN Korea) study investigated the characteristics, quitting behaviors, and biomarkers of new tobacco products and related products users in South Korea, funded by Korea Centers for Disease Control and Prevention. Considering the low prevalence of some groups in the general population (e.g., exclusive e-cigarette users), we targeted a convenience sample of individuals aged 19 years or older from March 2019 to June 2019. We categorized the product users into seven groups according to the combinations of each product they use (exclusive users (CC, e-cigarette, and HTP), dual users (CC + e-cigarette, e-cigarette + HTP, and CC + HTP), and triple users (CC + e-cigarette + HTP)). For effective analysis, a minimum sampling number was assigned to 300 adults per group. A survey was conducted using a structured questionnaire in two ways. First, an online survey was conducted using a sample from a panel managed by a major Korean research agency, Gallup Korea (https://www.gallup.co.kr/), that comprised 1.1 million members as of February 2019. Second, we conducted another face-to-face interview using tablet-assisted personal interviewing at a health checkup center and university. Participants were initially screened to be organized into each group through a series of questions about each of following three types of tobacco products and related products: CC, e-cigarette, and HTP. As part of the preamble on tobacco use, the options were characterized by detailed descriptions and pictures of e-cigarettes and HTPs to prevent confusion with other products. Given the official name of the respective HTP, we added the specific brand name of HTPs on sale in South Korea. After being placed into one of the eight groups, they were given a detailed questionnaire. All participants received financial incentives equivalent to 3000 Korean Won (KRW = 0.00084 USD) for participating in the online panel or KRW 10,000 for offline participation. This study received approval from the Institutional Review Board of Soonchunhyang University (201811-BR-046-03).

### 2.2. Measures

#### 2.2.1. Types of Tobacco Products and Related Products Use and Behaviors

The cigarette smoking was assessed by the following question: “Do you currently smoke cigarettes every day, some days, or not at all”. Current cigarette smokers were those who responded as every day or some days with lifetime use of 100 or more cigarettes. E-cigarette use was assessed by asking: “Have you ever used e-cigarettes in your life?” (yes/no) and “Did you use e-cigarettes in the past 30 days?” (yes/no). Current e-cigarette users were those who had used e-cigarettes in their lifetime and the past 30 days. HTP use was assessed through the following question: “Have you ever used heated tobacco products (e.g., IQOS, Glo, Lil) in your life?” and “Do you use currently heated tobacco products every day, some days, or not at all in the past 30 days?” Current HTP users were those who responded as every day or some days in the past 30 days.

Detailed behaviors related to tobacco use were also assessed. The duration of use was analyzed by year, and the amount of CC and HTP use was assessed by cigarettes/sticks smoked per day. Moreover, the amount of e-cigarette use was assessed by puffing sessions in a day, and nicotine dependence was calculated by the modified Heaviness of Smoking Index (HSI) [14]. For e-cigarette users without nicotine content in their solution, we set the nicotine dependence to 0.

Attempts to quit each product were asked separately and quit attempts were defined as each user who answered “yes” to the question: “In the past 12 months, have you attempted to quit CCs/e-cigarettes/HTPs?” For dual or triple users, two or three questions about their attempts to abstain from each product were given repeatedly according to their current status for tobacco use. The readiness to quit each product was categorized, according to the transtheoretical model, into pre-contemplation, contemplation, and preparation stage. [15] For all participants, perceived harmfulness of e-cigarette or HTP was assessed by, “Do you think e-cigarettes (or heated tobacco products) are more harmful than regular cigarettes, less harmful, or are they equally harmful to health?”

#### 2.2.2. Covariates

Demographic characteristics, including sex, age, educational attainment, household income, and marital status, were collected. Age was categorized as follows: <30, 30–39, 40–49, or ≥50 years. Education was categorized as high school level or lower, college level, or postgraduate level. Household income was categorized as KRW <3,000,000, 3,000,000–4,999,999, or ≥5,000,000. Marital status was categorized as married, never married, divorced or separated, or widowed. Whether there were any adolescents at home or not was also asked. Alcohol consumption frequency was categorized as follows: <1/month, 2–4/month, or at least weekly. Current use of medication for hypertension, diabetes mellitus, or dyslipidemia and past diagnosis of coronary heart disease, cerebrovascular disease, or cancer were asked. Subjective health was categorized as either good, fair, or bad, and presence of chronic cough in the past 3 months was asked (yes/no).

### 2.3. Statistical Analysis

Descriptive statistics were presented as numbers and percentages depicting sociodemographic characteristics. Additionally, tobacco use behaviors (frequency, amount, duration, time to first use, and modified HSI) were presented according to the type of tobacco products and related products. Chi-square tests with Bonferroni correction for multiple comparisons were conducted to compare their perceived harm of HTP use (i.e., whether or not they used HTPs), and McNemar tests were conducted to compare proportions of quit attempts among triple users. To compare the quit attempts using the product, multivariate Poisson regression analysis was conducted among exclusive users. Compared with the quit attempts among exclusive cigarette smokers, adjusted prevalence ratio (aPR) and 95% confidence interval (CI) were calculated for exclusive e-cigarette or HTP users with adjustment for possible confounders. Similarly, the association between readiness to quit and the type of products among exclusive users were examined using multivariate analysis. Finally, for cigarette smokers (exclusive cigarette smokers, dual users with CC and e-cigarette, dual users with CC and HTP, or triple users), the association between poly-use and quit attempts was examined to decide whether poly-use was related to increased or decreased quit attempts. We conducted the same analysis for e-cigarette users and HTP users, respectively. STATA 14.0 (College Station, TX, USA) was used for statistical analysis, and *p*-values < 0.05 were considered statistically significant. 

## 3. Results

### 3.1. General Characteristics and Tobacco Use Behaviors

The general characteristics of 2831 respondents included in the final analysis are summarized in Table 1. The number of subjects according to the type of tobacco products and related products is 725 (25.6%) exclusive cigarette smokers, 316 (11.2%) exclusive e-cigarette users, 377 (13.3%) HTP users, 374 (13.2%) dual users with CC and e-cigarette, 303 (10.7%) dual users with e-cigarette and HTP, 393 (13.9%) dual users with CC and HTP, and 343 (12.1%) triple users. The vast majority of respondents were male (78.2%), residents of the metropolis (63.4%), or city (34.4%), and their educational attainment was above college (82.0%). Those taking medications regularly with comorbidity of hypertension, dyslipidemia, and diabetes, and those with a history of any cancer, coronary artery disease, and cerebrovascular disease were 17.6% and 5.9% of respondents, respectively. Less than 10% of respondents reported symptoms of chronic cough of more than 3 months. Respondents’ tobacco use behavior according to the type of tobacco products and related products is summarized in Table 2. Moreover, 75.4% of cigarette smokers, 57.6% of HTP users, and 36.0% of e-cigarette users utilized the products daily. The mean duration of using the products was 19.5 years for CCs, 1.58 years for HTPs, and 1.93 years for e-cigarettes.

### 3.2. Past Year Quit Attempts According to the Type of Tobacco Products and Related Products

Triple users attempted to quit CCs significantly more than e-cigarettes or HTPs in the past 1 year (*p* < 0.001, respectively) according to the McNemar test, while there was no difference in quitting attempts between e-cigarettes and HTPs. Multivariate Poisson regression analysis after adjustment for variables, such as age, sex, education level, household income, marital status, children at home, alcohol consumption frequency, comorbidity of hypertension, dyslipidemia, and diabetes confirmed by medications, history of cancer, coronary artery disease, or cerebrovascular disease, chronic cough of more than 3 months, self-rated health status, and depressive mood in the past 2 weeks, showed that dual users with CC and e-cigarette attempted to quit CCs more than exclusive cigarette smokers in the past 1 year, with marginal significance (aPR = 1.20, 95% CI 1.00–1.44, *p* = 0.044). Triple users with CC, e-cigarette, and HTP also attempted to quit CCs more than exclusive cigarette smokers in the past 1 year (aPR = 1.37, 95% CI 1.14–1.65, *p* = 0.001). There was no significant difference in the rate of attempts to quit e-cigarettes among dual or triple users, compared with exclusive e-cigarette users in the past 1 year. However, compared with exclusive HTP users, dual users with e-cigarette and HTP or triple users have a significantly higher percentage of HTP quitting attempts in the past 1 year (aPR = 1.73, 95% CI 1.35–2.22, aPR = 1.32, 95% CI 1.02–1.70, respectively), while no significant difference was observed for this rate in dual users with CC and exclusive HTP users in the past 1 year. (Table 3).

In the subgroup analysis for daily users, triple users with CC, e-cigarette, and HTP attempted to quit CCs more than exclusive cigarette smokers in the past 1 year (aPR = 1.44, 95% CI 1.15–1.81, *p* = 0.002). Dual users with e-cigarettes and HTP (aPR = 1.74, 95% CI 1.02–2.99, *p* = 0.043) and triple users (aPR = 2.18, 95% CI 1.28–3.71, *p* = 0.004) attempted to quit e-cigarettes more than exclusive e-cigarette users in the past 1 year. Dual users with e-cigarettes and HTP (aPR = 1.89, 95% CI 1.29–2.76, *p* = 0.001) and triple users (aPR = 1.78, 95% CI 1.25–2.55, *p* = 0.02) attempted to quit HTPs more than exclusive HTP users in the past 1 year (data not shown).

### 3.3. Readiness to Quit and Past Year Quit Attempts among Exclusive Users

After adjusting the variables, multivariate Poisson regression analysis concerning the readiness to quit tobacco products and related products among exclusive users denoted that HTP users were less prepared to quit the products compared to cigarette smokers (aPR = 0.52, 95% CI 0.35–0.76, *p* = 0.001), while no difference of readiness to quit was observed between exclusive cigarette smokers and e-cigarette users. Multivariate Poisson regression analysis also highlighted that e-cigarette users and HTP users were less likely to attempt to quit the products compared with cigarette smokers among those who use any single product (aPR = 0.80, 95% CI 0.64–1.00, aPR = 0.59, 95% CI 0.47–0.74, respectively) (Table 4).

### 3.4. Relative Harm Perception Concerning Heated Tobacco Products

There was a significant difference in relative harm perception about HTP use among respondents. The proportion of having the belief that HTP were less harmful than CC was significantly higher among those who use only HTP, followed by those who use HTP and other products, and those who do not use HTP (44.8%, 32.4%, 17.9%, respectively, *p* < 0.017, using chi-square tests with Bonferroni correction for multiple comparisons). (Figure 1).

## 4. Discussion

The new tobacco products and related products, such as e-cigarettes and HTPs, were marketed as being less harmful than CCs or as helpful to stop smoking despite limited evidence. Due to this, the number of smokers using e-cigarettes or HTPs began to dramatically increase resulting in the rapid evolvement of this market [7] and renormalize tobacco use. The present study demonstrates the association between the pattern of the product use and the readiness to quit or attempts to quit the product among Korean adults. To our knowledge, this is the first report to investigate attempts to quit every type of product used by smokers. 

Unlike with e-cigarettes, our results showed that dual users with CCs and HTPs were less likely to attempt quitting both products than exclusive users of each product, and exclusive HTP users were less motivated to quit HTPs and less likely to attempt quitting the products than cigarette smokers. Similarly, a recent study concerning Korean adolescents disclosed that using e-cigarettes among current smokers was associated with a higher odd of cigarette quit attempts but using HTPs among current smokers was not associated with cigarette quit attempts [13]. Another study about Korean adolescents showed that the use of e-cigarettes and/or HTPs among ever-smokers of cigarettes was associated with lower odds of cigarette smoking cessation, although these studies did contain certain limitations that there was no data regarding quit attempts or abstinence from e-cigarettes or HTPs among the product users [6]. Japan’s experience also showed a similar trend that cigarette sales have likely been reduced through the rollout of IQOS, while combined product volume remained unchanged [16]. American Cancer Society suggested that HTPs are likely to be replacing cigarette sales in Japan [17]. Consistently, our results suggested the possibility that people who choose HTPs were less active in quitting tobacco. 

The relative harm perception of HTP is one of the independent predictors of initiation or continuation of use HTP, and recent studies from various countries suggested that HTP-including poly-users may have higher nicotine dependence and are likely to underestimate their harmful effects to a greater extent than exclusive users [6,18,19,20]. Interestingly, our results showed that the perception that HTP is less harmful than CC was highest among exclusive HTP users, followed by poly-users of HTPs and other products, and non-HTP users. Several studies have analyzed the potential harm emanating from HTPs, however, the nicotine levels of HTPs were 70–80%, similar to CCs and higher than e-cigarettes. One analysis of HTPs revealed that there were no significant differences in most biomarkers of potential harm between HTP users and cigarette smokers [21]. Other recent studies suggested possible hepatotoxicity of HTPs and a positive association between ever using HTPs and asthma, allergic rhinitis, and atopic dermatitis among adolescents [22,23]. 

If the marketing of HTPs in the US is focused only on harm reduction, the vast majority of smokers will be less motivated to quit or attempt to quit smoking while being satisfied with the use of HTPs, and the proportion of poly-users will increase. Another issue of using e-cigarettes or HTPs among current smokers is that many of them failed to quit smoking and rather became poly-users themselves. Poly-users have higher nicotine dependence, and it is difficult for them to achieve smoking cessation despite having higher motivation and attempts to quit smoking than cigarette smokers. Piper et al. showed that about half of dual users with CCs and e-cigarettes failed to quit smoking and continued to use both products, and only 5.9% switched from CCs to e-cigarettes completely and 1.4% abstained from both products after 1 year in a real-world setting [24]. Similar to previous results, the data from the Ministry of Health and Welfare in South Korea showed that 42.2% of cigarette smokers, 85.8% of e-cigarette users, and 86.6% of HTP users were dual or triple users of CCs, e-cigarettes, and/or HTPs [25].

The strengths of the current study are as follows. First, one of the challenges of the studies for e-cigarettes or HTPs was the difficulty to recruit enough participants because of the lower prevalence of using e-cigarettes and/or HTPs compared with CCs in South Korea. According to the Korean National Health and Nutrition Examination Survey data, the prevalence of current smokers was 36.7% among males and 7.5% among females, and the prevalence of e-cigarettes (6.3% among males and 0.9% among females) or HTPs (7.9% among males and 0.7% among females) use was much lower than cigarette smoking [26]. In particular, the proportion of exclusive e-cigarette users and dual use of e-cigarettes and HTPs is much lower than 0.2%. Therefore, like the current study, it is more reasonable to recruit sufficient participants for each type of product group equally by non-probability targeted sampling methods, rather than using representative samples [27]. Second, to increase the reliability of the study, we tried to classify tobacco products and related products users accurately by using a screening questionnaire that included pictures of products and a detailed description of specific brand names. In the meantime, the use of e-cigarettes or HTPs did not verified accurately in South Korea. HTPs were commonly confused with e-cigarettes since the Korean government categorizes HTPs as “cigarette-type e-cigarettes”, which was the same classification used for e-cigarettes [28]. A recent study in South Korea revealed that there was a discordance of about 40% between self-reported and interviewer-rated tobacco use patterns [29]. The studies in UK and US also reported that mentioning brand names of a specific product type or using pictures of products is likely to improve the accuracy of the assessment [30,31]. Third, we investigated the association between the pattern of product use and readiness to quit or quit attempts according to every type of product used by smokers, after adjusting for multiple potential confounders using data from a large-scale survey.

Due to the nature of this study, however, several limitations should be considered when interpreting the results. First, the participants in the current study were recruited by using non-probability targeted sampling methods because of the low prevalence of using e-cigarettes and/or HTPs in South Korea. Therefore, the samples were not representative of the general population and nationally representative surveys are needed to complement the data. Similar to previous studies regarding new tobacco products and related products, there are possibilities for recruiting a higher income, literate, younger, and more male population in the current study [32,33]. Second, causality, reverse causality, and temporal relationships could not be ascertained based on cross-sectional data. There is a possibility that smokers who are interested in quitting smoking are more likely to try using e-cigarettes or HTPs and more likely to become dual or triple users eventually. In other words, exclusive users try relatively little to quit smoking. Our results suggest that exclusive HTP users are less likely to try quitting smoking after switching from cigarette smokers to exclusive HTP users. Further studies using a prospective randomized controlled trial will be needed to verify the causation of our findings. Third, data were collected via self-reported surveys and, thus, might have been subject to recall bias and underestimation. However, we tried to minimize the recall bias among participants by using a screening questionnaire that included pictures of products and a detailed description of specific brand names. Finally, we may not have fully accounted for potential confounders in the analysis. However, we tried to adjust for smoking-related factors, such as socioeconomic status, level of nicotine dependence, chronic cough, frequency of alcohol consumption frequency, smoking-related comorbidities, and psychosocial risk factors.

## 5. Conclusions

In the past 1-year, triple users attempted to quit CCs more than exclusive cigarette smokers. Additionally, triple users and dual users with e-cigarette and HTP were more inclined to quit HTPs than exclusive HTP users. Furthermore, HTP users were less prepared to quit the product compared with cigarette smokers, and exclusive HTP users were more likely to believe that HTPs were less harmful than CCs. These findings could explain the attenuation of trends in smoking cessation recently witnessed in South Korea. More campaigns are needed to give smokers the right perception of HTPs or e-cigarettes and encourage them to quit using all tobacco products and related products completely. 

## Figures and Tables

**Figure 1 ijerph-17-08622-f001:**
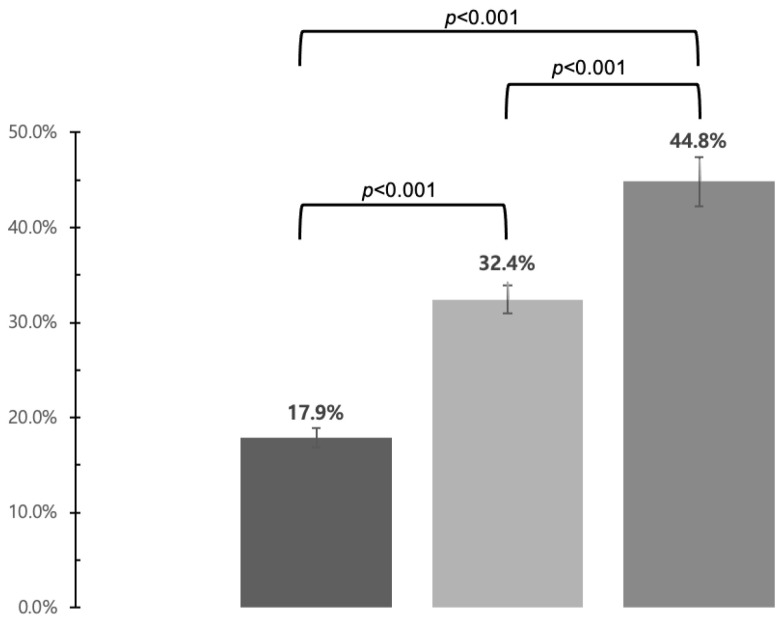
Proportion of those who answered that heated tobacco products are less harmful than conventional cigarettes. Three groups were compared in pairs using chi-square tests with Bonferroni correction for multiple comparison (*p* < 0.017). Abbreviation: HTP, heated tobacco product.

**Table 1 ijerph-17-08622-t001:** General characteristics of study subjects (*n* = 2831).

Characteristics		*n*	%
Age (year)	≤29	612	21.6
	30~39	754	26.6
	40~49	802	28.3
	≥50	663	23.4
Sex	Male	2213	78.2
	Female	618	21.8
Survey mode	Off-line	942	33.3
	On-line	1889	66.7
Residential area	Metropolitan	1796	63.4
	Another city	973	34.4
	Rural area	62	2.2
Education level	≤High school	509	18
	College	1974	69.7
	≥Postgraduate	348	12.3
Household income (KW)	<300	425	15
	300–499	915	32.3
	≥500	1491	52.7
Marital status	Married	1606	56.7
	Never married	1152	40.7
	Divorced or separated	62	2.2
	Widowed	11	0.4
Children at home	Yes	1252	44.2
	No	1579	55.8
Type of tobacco products and related products use	Exclusive CC	725	25.6
	Exclusive EC	316	11.2
	Exclusive HTP	377	13.3
	Dual (CC + EC)	374	13.2
	Dual (EC + HTP)	303	10.7
	Dual (CC + HTP)	393	13.9
	Triple	343	12.1
Alcohol consumption frequency	≤1/month	804	28.4
	2~4/month	999	35.3
	At least weekly	1028	36.3
Medication for HTN/DM/DL	Yes	499	17.6
Diagnosis of CHD/CVD/cancer	Yes	166	5.9
Chronic cough	Yes	276	9.8
Subjective health	Good	880	31.1
	Fair	1598	56.5
	Bad	353	12.5
Depressive mood	Yes	443	15.7

Descriptive statistics were presented as numbers (*n*) and percentages (%). Abbreviations: CC, conventional cigarette; EC, electronic cigarette; HTP, heated tobacco product; HTN, hypertension; DM, diabetes mellitus; DL, dyslipidemia; CHD, coronary heart disease; CVD, cerebrovascular disease.

**Table 2 ijerph-17-08622-t002:** Tobacco use characteristics.

Smoking Characteristics (*n* = 1835)		*n* (%) or Mean ± SD
Frequency	Daily	1383 (75.4)
	Intermittent	452 (24.6)
Cigarettes per day	≤10	1102 (60.1)
	11~20	652 (35.5)
	21~30	67 (3.6)
	≥31	14 (0.8)
Time to first cigarette (min)	>60	551 (30.0)
	31~60	353 (19.2)
	6~30	635 (34.6)
	≤5	296 (16.1)
HSI (0~6 point)		1.82 ± 1.42
Duration of use (year)		19.51 ± 11.95
Quit cigarette smoking in the past year		1020 (55.6)
Readiness to quit smoking	Preparation	302 (16.5)
	Contemplation	409 (22.3)
	Precontemplation	1124 (61.3)
**Vaping characteristics (*n* = 1336)**		
Frequency of e-cigarette use	Daily	481 (36.0)
	Intermittent	855 (64.0)
Nicotine concentration (mg/mL)	0 (nicotine-free)	85 (6.4)
	0.1~1.0	375 (28.1)
	1~3	352 (26.3)
	4~6	239 (17.9)
	≥7	100 (7.5)
	Don’t know	185 (13.8)
Sessions per day using e-cigarette	≤10	1180 (88.3)
	11~20	111 (8.3)
	21~30	27 (2.0)
	≥31	18 (1.4)
Time to first e-cigarette (min)	>60	570 (42.7)
	31~60	210 (15.7)
	6~30	355 (26.6)
	≤5	201 (15.0)
HSI (0~6 point)		1.30 ± 1.31
Duration of use (year)		1.93 ± 1.90
Quit e-cigarette use in the past year		624 (46.7)
Readiness to quit e-cigarette use	Preparation	266 (19.9)
	Contemplation	314 (23.5)
	Precontemplation	756 (56.6)
**HTPs using characteristics (*n* = 1416)**		
Frequency of HTP use	Daily	815 (57.6)
	Intermittent	601 (42.4)
Times per day using HTP	≤10	1115 (78.8)
	11~20	271 (19.2)
	21~30	21 (1.5)
	≥31	8 (0.6)
Time to first stick (min)	>60	579 (40.9)
	31~60	291 (20.6)
	6~30	400 (28.3)
	≤5	146 (10.3)
HSI (0~6 point)		1.32 ± 1.30
Duration of use (year)		1.58 ± 1.50
Quit HTP use in the past year		561 (39.6)
Readiness to quit HTP use	Preparation	191 (13.5)
	Contemplation	320 (22.6)
	Precontemplation	905 (63.9)

Descriptive statistics were presented as numbers (*n*) and percentages (%) or mean ± standard deviation. Abbreviations: SD, standard deviation; HSI, Heaviness of Smoking Index; HTP, heated tobacco product.

**Table 3 ijerph-17-08622-t003:** Multivariate association between the type of tobacco products and related products use with quit attempts in the past year.

			Crude			Multi-Adjusted ^1^		
	*n*	%	PR	95% CI	*p*	PR	95% CI	*p*
**Quit cigarette smoking among current smokers in the past year (*n* = 1835)**								
Exclusive CC	725	46.9	Ref.			Ref.		
CC + E-cigarette	374	58.8	1.25	1.06–1.49	0.009	1.20	1.00–1.44	0.044
CC + HTP	393	56.2	1.20	1.01–1.42	0.036	1.16	0.96–1.38	0.116
Triple	343	69.7	1.49	1.26–1.75	<0.001	1.37	1.14–1.65	0.001
**Quit e-cigarette use among e-cigarette users in the past year (*n* = 1336)**								
Exclusive e-cigarette	316	40.8	Ref.			Ref.		
CC + E-cigarette	374	38.0	0.93	0.73–1.18	0.551	0.86	0.67–1.09	0.215
E-cigarette + HTP	303	62.7	1.54	1.23–1.92	<0.001	1.22	0.97–1.54	0.092
Triple	343	47.5	1.16	0.92–1.47	0.197	0.96	0.75–1.22	0.729
**Quit HTP use among HTP users in the past year (*n* = 1416)**								
Exclusive HTP	377	29.4	Ref.			Ref.		
E-cigarette + HTP	303	60.4	2.05	1.62–2.60	<0.001	1.73	1.35–2.22	<0.001
CC + HTP	393	28.5	0.97	0.74–1.26	0.808	0.97	0.74–1.26	0.794
Triple	343	45.2	1.53	1.20–1.96	0.001	1.32	1.02–1.70	0.033

^1^ Adjusted for age, sex, education level, household income, marital status, children at home, alcohol consumption frequency, current use of medication for hypertension/diabetes mellitus/dyslipidemia, past diagnosis of coronary heart disease/cerebrovascular disease/cancer, chronic cough for 3 months, self-rated health status, and depressive mood in the past 2 weeks. Abbreviations: PR, prevalence ratio; CI, confidence interval; CC, conventional cigarette; E-cigarette, electronic cigarette; HTP, heated tobacco product.

**Table 4 ijerph-17-08622-t004:** Multivariate association between exclusive use of tobacco products and related products and readiness to quit and quit attempts in the past year (*n* = 1418).

			Crude			Multi-Adjusted ^1^		
	*n*	%	PR	95% CI	*p*	PR	95% CI	*p*
**Readiness to quit the tobacco products and related products (preparation stage)**								
Exclusive CC	725	19.2	Ref.			Ref.		
Exclusive e-cigarette	316	17.4	0.91	0.66–1.24	0.544	0.93	0.66–1.31	0.685
Exclusive HTP	377	10.1	0.53	0.37–0.75	<0.001	0.52	0.35–0.76	0.001
**Quit attempts using the tobacco products and related products in the past year**								
Exclusive CC	725	46.9	Ref.			Ref.		
Exclusive e-cigarette	316	40.8	0.87	0.71–1.07	0.18	0.80	0.64–1.00	0.046
Exclusive HTP	377	29.4	0.63	0.51–0.78	<0.001	0.59	0.47–0.74	<0.001

^1^ Adjusted for age, sex, education level, household income, marital status, children at home, alcohol consumption frequency, current use of medication for hypertension/diabetes mellitus/dyslipidemia, past diagnosis of coronary heart disease/cerebrovascular disease/cancer, chronic cough for 3 months, self-rated health status, and depressive mood in the past 2 weeks. Abbreviations: PR, prevalence ratio; CI, confidence interval; CC, conventional cigarette; E-cigarette, electronic cigarette; HTP, heated tobacco product.

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
