# Peer review of "Are Heated Tobacco Product Users Less Likely to Quit than Cigarette Smokers? Findings from THINK (Tobacco and Health IN Korea) Study"

_ijerph, 2020, doi:10.3390/ijerph17228622_

Round 1

Reviewer 1 Report

Review

IJERPH

Are heated tobacco product users less likely to quite cigarettes smokers ?

Sur le fonds

This paper from South Korea concerns a cross-sectional study of 2,831 Korean smokers and vapers and studies smoking cessation among those who smoke regular cigarettes, heated tobacco or use vape.

We note that the consumption of heated tobacco is important in Korea (a word about the promotion of HTP and its taxation in Korea would be welcome).

The trajectory of users is not described in this cross-sectional study. Also, it is difficult to conclude on a causal link between the products consumed and the judgment.

Thus, for example, it can be anticipated that smokers who take up e-cigarettes do so to quit smoking. Those who succeed are no longer for a time that exclusive vapers, those who continue to smoke and to be dual users, are already selected for their inability to stop smoking, creating a major bias and must be taken fully in the interpretation.

To note:

  • A tobacco product is a product that contains tobacco. The e-cigarette does not contain tobacco -> if we want to talk about the e-cigarette and put it in the same category, we must say each time "tobacco products and related products".
  • Nothing is said for the nicotine in e-cigarettes. The delivery of nicotine with the electronic cigarette is very regular over the day (like a nicotine patch) allowing to gradually reduce the needs of the nicotine-dependent smoker's body and does not produce nicotine peaks like cigarettes and http which reinforces the nicotine addiction.
  • Nothing is said that how many nicotine concentrations use nicotine-free e-cigarettes.

In details

Line 47: The tobacco industry's claim that heated tobacco reduces risk is well established in the laboratory but has not been demonstrated in real life. This claimed risk reduction must consider changes in smoking patterns to determine a possible risk reduction in real life.

Line 50 : FDA decisions on pneumonia relate to the use of cannabis with vitamin D (oily) that has nothing to do with normal e-cigarette use.
For cigarettes, for example, only tobacco cigarettes are taken into account, not cigarettes made from marijuana or from a mixture of tobacco and other drugs. àExplain that vaping accidents are not related to "tobacco products or related products" but to misuse using other products unrelated to tobacco.

Line 54: It is interesting to note the slowing down in smoking with the introduction of heated tobacco. It would have been important to give an idea of the decline in previous years and to say a word somewhere about the promotion of heated tobacco and its use by the tobacco industry to renormalize tobacco use in general.

Line 67: It would be interesting to have information on the use of heated tobacco among never smokers, to assess whether heated tobacco becomes a product of entry into tobacco smoking use.  Is such information available in Korea, as published in Japan (doi: 10.1136 / tobaccocontrol-2017-053947)

Line 105: The quantitative use of e-cigarette was better assessed by volume of e-liquid use each day (or the duration of a refill bootle) than with the number of puffs.

Line 107: The e-cigarette without nicotine a “tobacco product or a tobacco related product”? Have you some difference of cessation according to the use or not of nicotine in the e-liquid?

Line 166 table 2: There is an urge difference between daily users:

  • 36% for e-cigarette users,
  • 57,6% for HTTP users
  • 75,4% for cigarettes users.

Did you explore the quitting rate in daily users (compare with non-daily users of each product? On health point of view daily users are of main concern, occasional users are of less concern.

Line 179: OK for the demonstration that smokers who use electronic cigarettes combined with traditional cigarettes stop smoking more often, but
- is it because they had both products that they stopped smoking?

        or

- is it because a cigarette smoker who wants to quit tends to take the e-cigarette with a view to quitting?

Line 183: Cigarette smokers who wish to quit or smoke less sometimes use HTP and often e-cigarettes, while those who do not want to stay under one cigarette. The results observed are probably just as much a reflection of the patient's will or not to stop as the reflection of the effectiveness of one or the other of the associations. It is necessary here to be factual on the existence of the association and in the discussion to elaborate on the causality of these facts.

Line 234: Only a minority of HTP users believe that heated tobacco is less dangerous than cigarettes (and therefore do not follow the claims of tobacco companies to reduce the risk.

Line 266: The proportion of dual and triple users is extremely high in this Korean study, but it includes very intermittent users in the definition, for example 2/3 of users do not use it every day. It would have been interesting to have the data considering only the daily users.

Line 307 vous annoncez que les doubles et tripes users s’arrêtent plus d(utiliser un produit du tabac ou apparenté avec un lin causal, mais le lien inverse est tout aussi voire plus plausible ! un fumeur qui veut s’arrêter de fumer (ou réduire le risque) ajoutent la cigarette électronique ou le http pour arrêter de fumer. Ceux qui ajoutent la cigarette électronique arrêt plus, ceux qui ajoute l’http arrêt moins. Tout se passe comme si l’http était utilisé pour fumer autrement e la e-cigarette pour arrêter le tabac.

Line 234: Only a minority of HTP users believe that heated tobacco is less dangerous than cigarettes (and therefore do not follow the risk reduction claims of tobacco companies.

Line 266: The proportion of dual and triple users is extremely high in this Korean study, but it includes in the definition very intermittent users, for example 2/3 of e-cigarette users do not use it every day. It would have been interesting to have the data considering only the daily users.

Line 307 you report that double and guts users no longer stop using a tobacco or related product with a causal link, but the reverse link is just as or more plausible! a smoker who wants to quit (or reduce the risk) add e-cigarettes or http to quit smoking. Those who add e-cigarettes quit more, those who add http quit less. It is as if the HTP was used to smoke otherwise and the e-cigarette to quit smoking.

Reviewer 2 Report

The manuscript entitled "Are heated tobacco product users less likely to quit 3 than cigarette smokers? Findings from THINK 4 (Tobacco and Health IN Korea) Study" is considered to be an interesting attempt to evaluate the cigarette smokers preference for available mode of smoking.

The concept is new and well reported here. The strength of the study is sample size. Since sample size is high, the results show significant changes between groups. The methods are well defined. The financial status is one of the main criteria to choose the mode of cigarette smoking.

I appreciate the authors, carefully selecting one of the criteria as household income in their questions to determine the outcome. Authors already wrote clear limitations of the study. The data are well discussed. It would be better to include inclusion and exclusion criteria for this study.  

I recommend to accept the current format.

Thank you

Author Response

We would like to thank you for considering our manuscript entitled “Are heated tobacco product users less likely to quit than cigarette smokers? Findings from THINK (Tobacco and Health IN Korea) Study” for publication in “International Journal of Environmental Research and Public Health”. We also thank the reviewers for their constructive suggestions.

Reviewer 3 Report

In this article the authors performed a cross-sectional study among adult tobacco users to examine the association between the use of new tobacco products (such as e-cigarettes and heated tobacco products) and cessation behaviors. The new tobacco products are marketed as “less harmful than cigarettes” or “helpful to stop smoking”, and this is reflected in the results, since heat tobacco product users were less likely to quit than cigarette smokers, with the perception that heat tobacco products are less harmful (although published evidence indicates otherwise). All results were adjusted to covariates (age, sex, education level, household income, marital status, children at home, etc).

The study is interesting and well performed. I will just ask the authors to include all questionnaires in English as supplementary files, and complete marital status in Table 1. "Marital status" included several categories as indicated in lines 122-123, however Table 1 only shows one category; marital status could be relevant in tobacco use and willingness to quit (mainly if children are present).

Please discuss further and indicate in the Conclusions section whether campaigns should be implemented to inform smokers about the potential dangers of e-cigarettes and heat tobacco products; this is important because new tobacco products give them a false perception of security.  
